# Synthesis, Characterization and Antimicrobial Studies of Ti-40Nb-10Ag Implant Biomaterials

**Bin Zhu [1], Yuqin Zhang [1], Yongcheng Chen [2], Ping Yuan [2], Wentong Wang [2], Hao Duan [3,*] and Zhihua Wang [2,*]**

[1] School of Materials Science & Engineering, Kunming University of Science and Technology, Kunming 650093, China
[2] Trauma Center, The First Affiliated Hospital of Kunming Medical University, Kunming 650000, China
[3] Department of Sports Medicine, The First Affiliated Hospital of Kunming Medical University, Kunming 650000, China
[*] Correspondence: kmu.dh@hotmail.com (H.D.); wzh3333@126.com (Z.W.)

**Abstract:** Bacterial infection and stress shielding are important issues in orthopedic implants. In this study, Ag element was selected as an antibacterial agent to develop an antibacterial Ti-40Nb-10Ag alloy by spark plasma sintering (SPS). The microstructure, phase constitution, mechanical properties, microhardness, and antibacterial properties of the Ti-40Nb-10Ag sintered alloys with different sintering temperatures were systematically studied by X-ray diffraction (XRD), scanning electron microscope (SEM), microhardness tests, compressive tests, and antibacterial tests. The Ti-40Nb-10Ag alloys were mainly composed of α-Ti, β-Ti, and Ti2Ag intermetallic phases. This study shows that the change in sintering temperature affects the microstructure of the alloy, which results in changes in its microhardness, compressive strength, elastic modulus, and antibacterial properties. At the sintering temperature of 975 °C, good metallurgical bonding was developed on the surface of the alloy, which led to excellent microhardness, compressive strength, elastic modulus, and antibacterial ability with an antibacterial rate of 95.6%. In conclusion, the Ti-40Nb-10Ag alloy prepared by SPS at 975 °C is ideal and effective for orthopedic implant.

**Keywords:** titanium alloy; low elastic modulus; high strength; antibacterial ability





## 1. Introduction

Ti-40Nb-based shape memory alloys have received increasing attention due to their superior biocompatibility, nontoxic elements, high mechanical strengths-to-weight ratio, low density, low elastic modulus (reduction of stress shielding effects), and excellent shape memory properties. They have been widely used as novel Al-, V-, and Ni-free biomaterials for orthopedic and dental implants to avoid the potential risk of Al, V, and Ni hypersensitivity [1–3]. Among the Ti-40Nb based materials, superelastic behavior caused by the stress-induced martensitic transformation between cubic β phase and orthorhombic α″ martensite phase is observed in the Ti-(25.5–27) at. % Nb alloy [4,5]. The superelasticity of the recovered largest strain is presented in the Ti-26 at. % Nb alloy. This leads to its high mechanical strength and low elastic modulus (70 GPa) [6].

Ti-40Nb beta titanium alloy shows good mechanical strength and low elastic modulus [7], which are significant to bone implantation materials. However, most implants do not have antibacterial abilities, which can lead to the possibility of infection [8]. Postoperative infection is a major complication in orthopedic surgery; it has been reported that the bacterial infection rate could up to 30% after internal fixation. In addition, infections interfere with the combination of bone and implants, which might cause implantation failure [9,10]. Long term antibiotic treatment was required in clinic infection measurement, but prolonged use of antibiotics often generated antimicrobial resistance [11]. Therefore, implantation materials with good antibacterial properties are critical in preventing bacterial infection [10].

In order to impart antibacterial properties to metal materials, one of the most common approaches is to add elements with antibacterial capabilities, such as Cu and Ag [12–14], which can provide a broad antibacterial spectrum, high antibacterial efficiency, and biocompatibility. Silver has functions in a broad spectrum of antibacterial activities and is used in antibacterial implantation materials [15]. Shan et al. [9] proved that Ti-15 wt.%Ag alloy showed a high antibacterial rate when cultured in *E. coli*, as well as good biocompatibility. Mian [16] et al. proved that Ti-Ag alloys with different Ag concentrations had antibacterial properties. Although the antibacterial alloys of currently studied biomaterials such as Ti-Cu and Ti-Ag [10] have good antibacterial abilities, there are also problems such as high elastic modulus associated with them. In clinical practice, studies show that the "stress shielding" [17,18] effect occurs if the elastic modulus of the biomaterials do not match that of the natural bone. This effect results in the occurrence of bone atrophy or insufficient remodeling, which ultimately leads to failure of implantation. Therefore, it is necessary to introduce the antibacterial spectrum of Ag into the Ti-40Nb alloy with high strength, low elastic modulus, and high wear resistance. Spark plasma sintering (SPS) has the advantages of low sintering temperature, short sintering time, high density, and a clean preparation process. In this study, a Ti-40Nb-10Ag alloy was proposed and fabricated by a spark plasma sintering (SPS) process to enrich its antibacterial and mechanical properties. The effect of SPS temperature on the antibacterial properties, biocompatibility, and mechanical properties were investigated. Such properties include: chemical composition, microstructure and phase constitute, elastic property, microhardness, and antibacterial activity. The controlling antibacterial mechanism of this alloy was also studied to show that the alloy proposed by this study is an excellent candidate for orthopedic implants.

## 2. Materials and Methods

### 2.1. Preparation and Fabrication of Sample

High purity titanium particles (purity 99.8%, average particle size (APS)~30 μm), high purity niobium particles (purity 99.8%, APS~30 μm), and high purity silver particles (purity 99.8%, average particle size~48 μm) were used to prepare Ti-40Nb-10Ag alloy (The SEM images of the raw particles are shown in Figure 1).

The titanium powder, niobium powder, and silver powder were weighed (FA-200N, Shanghai Electronic Balance Co., Ltd., Shanghai, China) and then put into a stainless-steel ball mill jar with stainless steel grinding balls. The mass ratio of metal powders to stainless steel balls was 3 to 1. Before mechanical alloying, the jar with the metal powders and balls was vacuumed in order to minimize the oxidation of the titanium, niobium, and silver powders. The metal powders were alloyed in a planetary ball mill at a rotational speed of 320 rpm for 10 h. Then, the dried mixed powders were put into a graphite mold and the mold was put into spark plasma sintering equipment (SPS-515S, Syntex Inc., Kawasaki, Japan) for preparation. It was heated to the target temperature with a heating rate of 100 °C/min and kept for 8 min at constant temperatures of 950, 975, and 1000 °C. The as-sintered samples were then cooled in the furnace with a cooling rate of 50 °C/min to room temperature. During the sintering process, the axial pressure was continuously applied at 40 MPa, and the degree of vacuum was kept at 10 Pa.

To determine the approximate range of the sintering temperature, the transition temperature (Tg) of Ti-40Nb-10Ag and Ti-40Nb as-milled powders were measured by differential scanning calorimeter (DSC) at a heating rate of 10 K·min$^{-1}$ from room temperature to 1373 K in a crucible. From the DSC curves of Ti-40Nb and Ti-40Nb-10Ag alloy powders, it was found that the Ti-40Nb powder had an absorption peak at 1026.01 °C, and the Ti-40Nb-10Ag powder had an absorption peak at 967.75 °C (Figure 2). Thus, the sintering temperatures of Ti-40Nb-10Ag were selected to be 950 °C, 975 °C, and 1000 °C according to the peak absorption temperatures to explore the influence of different temperature-related parameters on the sintered samples. Specifically, Ti-40Nb alloy was prepared at a sintering temperature of 975 °C for comparison in order to study the effect of silver addition to the Ti-40Nb alloy. The process parameters of all samples are summarized in Table 1.

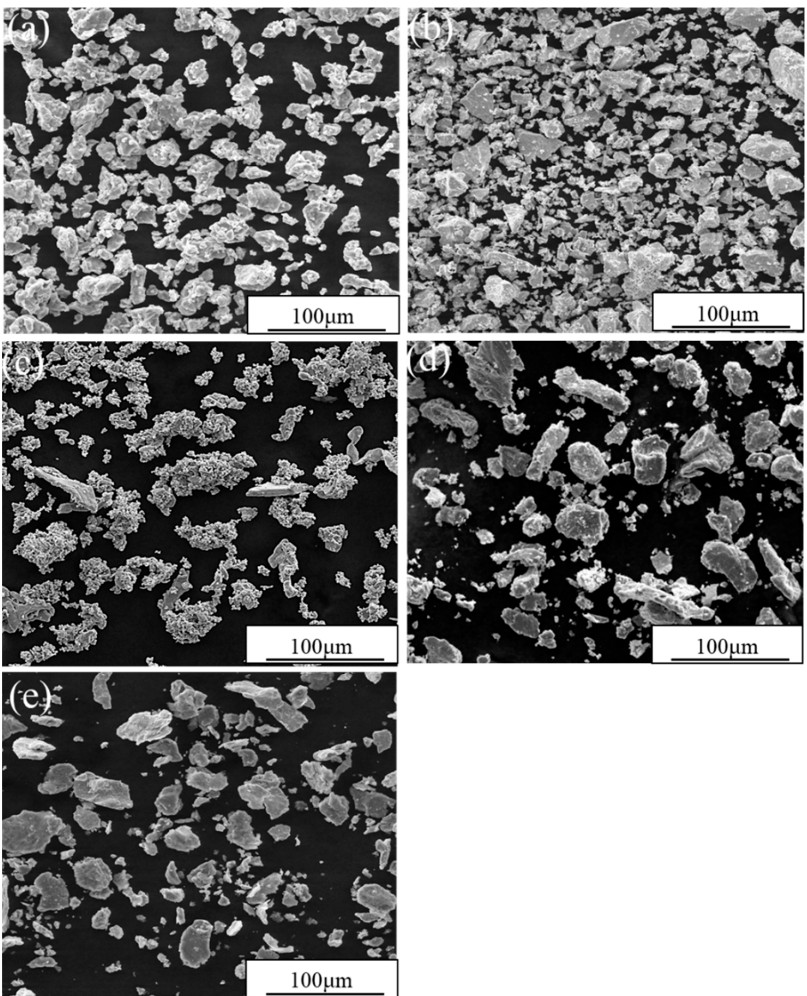

**Figure 1.** SEM images of primary metal particles (Ti, Nb, Ag)) and as-milled Ti-40Nb and Ti-40Nb-10Ag particles. (**a**) Ti; (**b**) Nb; (**c**) Ag; (**d**) Ti-40Nb; (**e**) Ti-40Nb-10Ag.

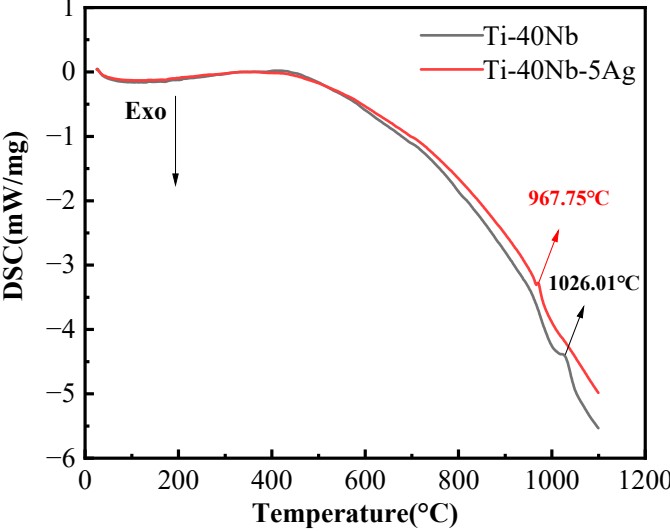

**Figure 2.** DSC curve of Ti-40Nb-10Ag and Ti-40Nb ball-milled alloy powders.

**Table 1.** The processing parameters of experimental samples.

| Sample | Alloy | Temperature (°C) |
|---|---|---|
| 1[#] | Ti-40Nb-10Ag | 950 |
| 2[#] | Ti-40Nb-10Ag | 975 |
| 3[#] | Ti-40Nb-10Ag | 1000 |
| 4[#] | Ti-40Nb | 975 |

### 2.2. Characterizations of Alloy

The phase composition of the alloy samples was characterized by the X-ray diffractometer (XRD) (D8 Advance, Bruker, Karlsruhe Baden-Württemberg, Germany) with Cu Kα radiation at 40 kV within diffraction angles ranging from 20° to 90° at a scanning speed of 2 degree·min$^{-1}$.

For SEM characterization, the fabricated alloy samples were cut into a shape of Φ 15 × 5 mm with electrical discharge wire-cut (EDWC), ground by sandpapers with different grit sizes (120[#], 240[#], 500[#], 800[#], 1000[#], 1200[#] in sequence), and polished with a polishing machine as the last step. The surface of the polished samples were cleaned with absolute ethanol, and the metallographic samples were etched with the prepared titanium alloy etching solution Kroll's (volume ratio of $HF:HNO_3:H_2O$ = 1:2:33). The microstructure of the samples was analyzed by SEM (Quanta 200, FEI, Waltham, MA, USA).

For mechanical property characterization, the fabricated alloy samples were cut into a shape of Φ 4 × 10 mm by EDWC and tested on a universal testing machine (Criterion 40, MTS, Eden Prairie, MN, USA) with a strain rate of $5 \times 10^{-4}$ s$^{-1}$ for compressive strength. These tests were done in triplicate to reduce experimental error and obtain accurate test results, and the samples were ground by SiC papers (200[#], 320[#], 400[#], 600[#] and 1000[#]) before the test. Samples were corroded for 40 s using Kroll's (volume ratio of $HF:HNO_3:H_2O$ = 1:2:33).

Microhardness was detected on the HMV-G21 (HMV-G21, SHIMADZU, Kyoto, Japan) microhardness equipment. The loading force was 200 g. The holding time of loading was 10 s. Three significantly different areas were chosen randomly and the results were averaged per those areas with standard deviation.

### 2.3. Antimicrobial Activity Tests

The fabricated alloy samples were cut into a shape of Φ 5 × 3 mm by EDWC, ground, polished, and sterilized. Gram-negative Escherichia coli (*E. coli*) (ATCC 8739) was used in the experiments. Briefly, 100 μL of the *E. coli* bacterial solution with a concentration of $10^7$ cell/mL was dispensed into a 50 mL centrifuge tube with the alloy. 10 mL of culture medium was then added into the tube. The bacteria with the culture medium and the alloys were cultured at 37 °C and 5% $CO_2$. After 24 h, the bacteria were diluted with PBS and 100 μL was taken out from the stock and spread on LB solid medium. The bacterial colonies were counted after culturing at 37 °C for 24 h. The antimicrobial rate of the test samples was calculated by the following equation.

$$\text{Antibacterial rate} = (N - M)/N \times 100\% \tag{1}$$

where N and M are the average number of the bacterial colonies on the Ti-40Nb and Ti-40Nb-10Ag with different sintering temperatures.

For the Ag concentration measurement, the ground, polished, and cleaned alloys were placed in a test tube containing 15 mL of 0.9 NaCl solution. The test tubes were immersed in a constant temperature water bath at 37 °C for 7 days. At 1, 3, 5, and 7 days, 5 mL of the solution was taken out, and the silver element in the solution was tested by inductively coupled plasma mass spectrometer (ICP-MS) (PerkinElmer NexION 300X, FEI, Waltham, MA, USA)

*2.4. Statistical Analysis*

The results were expressed as the mean ± standard deviation (SD); the independent experiment was duplicated at least three times. A one-way ANOVA was used to analyze the data. It was considered statistically significant when $p < 0.05$.

## 3. Results

*3.1. XRD*

Figure 3 shows the XRD patterns of Ti-40Nb-10Ag milled powders and Ti-40Nb-10Ag alloy with different sintering temperatures. The XRD pattern peaks for alpha Ti, pure Nb, and pure Ag was found in Ti-40Nb-10Ag milled powders. It should be noted that the XRD pattern peaks of beta Ti phase do not appear. This indicates that there was no mechanical alloying between Ti and Nb metal elements. From the results of the XRD patterns, three groups of samples with different sintering temperatures (1#,2#,3#) were composed of alpha Ti, beta Ti, and Ti$_2$Ag intermetallic phases.

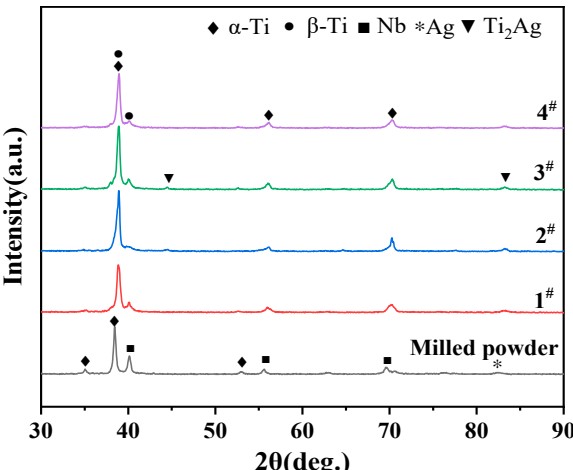

**Figure 3.** The XRD patterns of Ti-40Nb-10Ag milled powders and Ti-40Nb-10Ag alloy for different sintering temperature.

The diffusion of metal atoms under the direct current (DC) electric field is greatly promoted by the preparation of the SPS process. Similarly, the Joule heating of DC field is immediately emerged in the inter-particle voids by the high DC field energizing [6,19–21]. For that reason, it can accelerate the diffusion of Nb particles (β stabilizing element), which can lead to the formation of nearly single beta Ti with a small amount alpha Ti. In order to compare with Ti-40Nb-10Ag alloys, Ti-40Nb alloy (4#) was prepared at 975 °C. At this temperature, the alloy was dominated by single beta Ti with little alpha Ti phases.

*3.2. Microstructure*

Figure 4 shows the SEM images of the microstructure and morphologies of the Ti-40Nb-10Ag alloys with the different sintering temperatures. Figure 4d shows Ti-40Nb alloy at the sintering temperature of 975 °C. According to the SEM morphology of Figure 4a, due to the low sintering temperature, the diffusion between elements was not uneven and the elements were locally enriched. In the bulk element enrichment area, the darker area was enriched of light mass elements, while the whiter area was enriched of the heavy mass elements [22]. According to the Arrhenius equation:

$$K = A\exp\left(-\frac{Ea}{RT}\right) \tag{2}$$

where $K$ is the diffusion rate, A is the Arrhenius constant, Ea is the activation energy, R is the molar gas constant, and T is the thermodynamic temperature. With the heat fluctuations, the

probability of obtaining atoms with sufficient energy to diffuse across the barrier increases and the diffusion rate of the element increases with increasing temperature [23]. Secondly, a small number of holes can be observed. This is because, in the process of powder metallurgy sintering, the sintering neck between particles gradually grows up and the gap between particles gradually shrinks, which results in cracks or holes at the microscale.

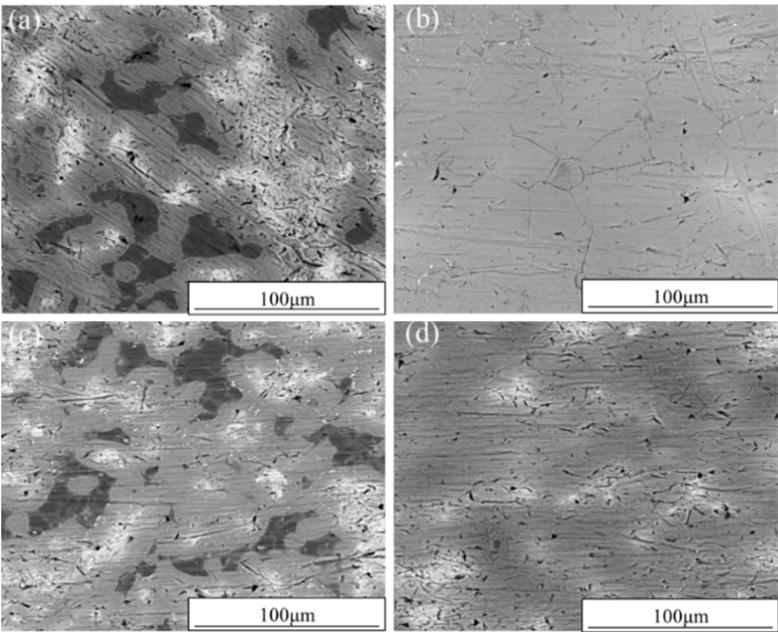

**Figure 4.** The SEM morphologies of Ti-40Nb-10Ag alloy for different sintering temperature. (**a**) 950 °C Ti-40Nb-10Ag; (**b**) 975 °C Ti-40Nb-10Ag; (**c**) 1000 °C Ti-40Nb-10Ag; (**d**) 950 °C Ti-40Nb.

When the sintering temperature (950 °C) was insufficient, the activation energy of the element sintering process was low, and the densification process of the alloy was incomplete. As a result, a large number of pores and defects in the alloy existed (Figure 4a). With the increase of sintering temperature (975 °C), the surface of 2# sample (Figure 4b) shows an excellent sintered morphology of powder metallurgy, with no element-enriched areas and less hole defects on the surface. The favorable sintered state was the basis for giving the material excellent mechanical properties, corrosion resistance, etc. [23]. During the sintering process with the higher sintering temperature, the higher plasma capacity was generated by the discharge. This resulted in powders obtaining more energy, which accelerated the movement between metal powders and produced a denser sintered-body alloy [24]. However, when the temperature was even higher (1000 °C), a large number of holes were observed on the surface of the 3# sample due to the high sintering temperature (Figure 4c). The holes mainly came from the Ag vacancies left by the diffusion and vaporization of the low-melting Ag. From the SEM surface morphology results in Figure 4a–c, it can be seen that an inappropriate sintering temperature would have adverse effects on the morphology of Ti-40Nb-10Ag alloy. When the sintering temperature was too low (Figure 4a), the alloy sample would produce an element-rich region; when the sintering temperature was too high (Figure 4c), the low-melting Ag in the alloy sample would volatilize and generate many pores. Figure 4d shows the sintered morphology of Ti-40Nb alloy at 975 °C sintering temperature, and the surface morphology of the sample showed good sintering results.

*3.3. Microhardness*

Figure 5 shows the microhardness of Ti-40Nb-10Ag alloys with different sintering temperatures. The microhardness of the alloy increased from 313.8 Hv (1#) to 360.4 Hv (2#), and then decreased to 338.4 Hv (3#). Combined with the results of surface morphology and microstructure, it showed that, when the sintering temperature was too low, there were many pores and defects in the alloy due to uneven diffusion of the element, which caused

the low microhardness of the alloy. When the sintering temperature increased, the surface of the alloy presented good powder metallurgy sintering morphology and no element enrichment area on the surface, as well as less hole defects. Its hardness was also improved. When the sintering temperature was too high, a large number of pores were formed on the surface of the alloy due to the volatilization of Ag with a low-melting point, which resulted in a hardness decrease. Compared with Ti-40Nb alloy, the addition of 10 wt.%Ag would significantly increase the microhardness of Ti-40Nb-10Ag due to the change in hardness value caused by the second phase in the alloy (Figure 3). Due to the particularity of spark plasma sintering, the second phase is distributed in the matrix in the form of flakes and some small particles. The degree of lattice distortion in the alloy increases, which results in a significant increase in the hardness of the alloy [25]. Microhardness values can be used as one of the important indicators to measure the wear resistance of the alloy; the higher of the microhardness, the better the wear resistance of the alloy was shown [26,27].

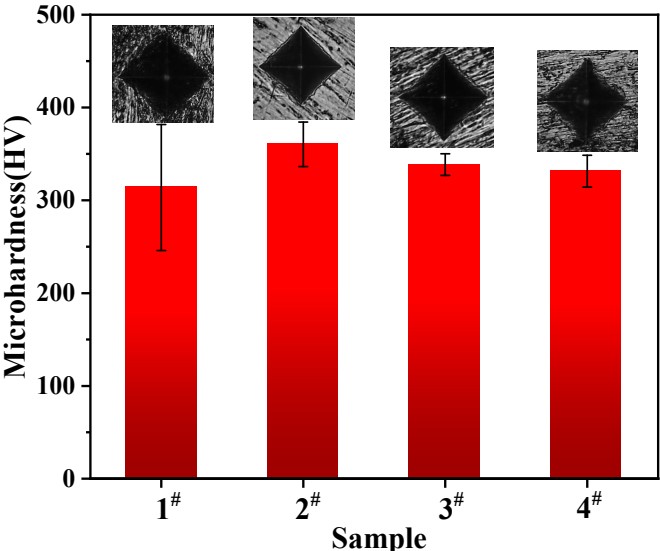

**Figure 5.** The microhardness of Ti-40Nb-10Ag for different sintering temperature (1#,2#,3#) and Ti-40Nb alloy (control group-4#).

### 3.4. Mechanical Properties

The Ti-40Nb-10Ag alloy is used for implant material in orthopedics, which is mainly subjected to the compressive stress after implantation in the body [1,28,29]. Therefore, the compressive performance tests of alloys are important to its functionality.

Figure 6 shows the results of compressive strength and elastic modulus of Ti-40Nb-10Ag alloy at different sintering temperatures. The compressive strength of the alloy increased from 1230 MPa (1#) to 1432 MPa (2#), and then decreased to 1193 MPa (3#). The change in the compressive strength of the alloy with the sintering temperature was closely related to the change in the microstructure. According to the SEM morphology of the alloy, when the sintering temperature was too low, the compressive strength of the alloy was low due to the existence of a large number of pores, defects, and uneven diffusion of elements. When the sintering temperature increases, the surface of the alloy presents good powder metallurgy sintering morphology (no element enrichment area, less hole defects) [6,30], which improves its compressive strength. When the sintering temperature was too high, due to the volatilization of Ag with a low melting point, a large number of pores were formed on the surface of the alloy and its compressive strength was reduced. Compared to Ti-40Nb alloy, the addition of 10 wt.%Ag would slightly reduce the compressive strength of Ti-40Nb-10Ag. The elastic modulus of the alloy increased from 59 GPa (1#) to 71 GPa (2#), and then decreased to 50 GPa (3#). The elastic modulus of the three samples was much lower than that of the TC4 alloy and was similar to that of natural human bone (10–30 GPa for cortical bone), which may not cause "stress shielding" [31] after implantation.

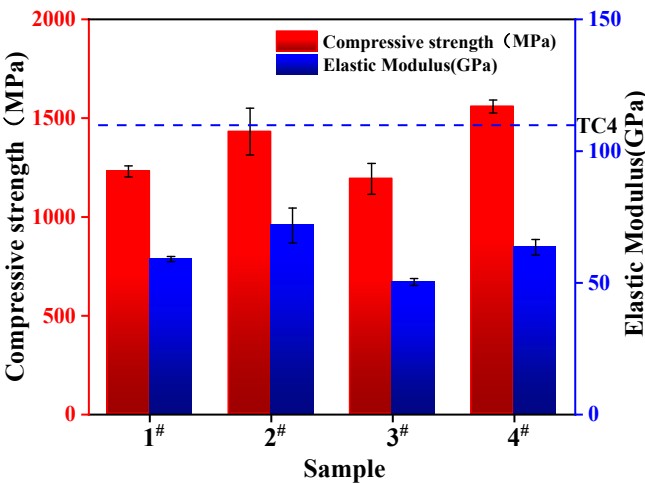

**Figure 6.** The compressive strengths and elastic modulus of Ti-40Nb-10Ag (1#, 2#, 3#) for different sintering temperatures and Ti-40Nb alloy (control group-4#).

### 3.5. Antibacterial Properties

Figure 7 shows typical *E. coli* bacterial colonies on Ti-40Nb-10Ag alloys of different sintering temperatures and Ti-40Nb after 24 h of incubation. Large numbers of bacteria colonies were found on the 4# sample, which confirmed that Ti-40Nb does not have an antibacterial property. For 1# and 3# sample, bacteria colonies were still observed, although the number of colonies was less than that on 4#. This suggested that Ti-40Nb-10Ag alloy fabricated under 950 °C and 1000 °C had limited antibacterial performance. For 2# sample, few colonies existed, which indicated that the Ti-40Nb-10Ag alloy fabricated under 975 °C showed a strong antibacterial ability.

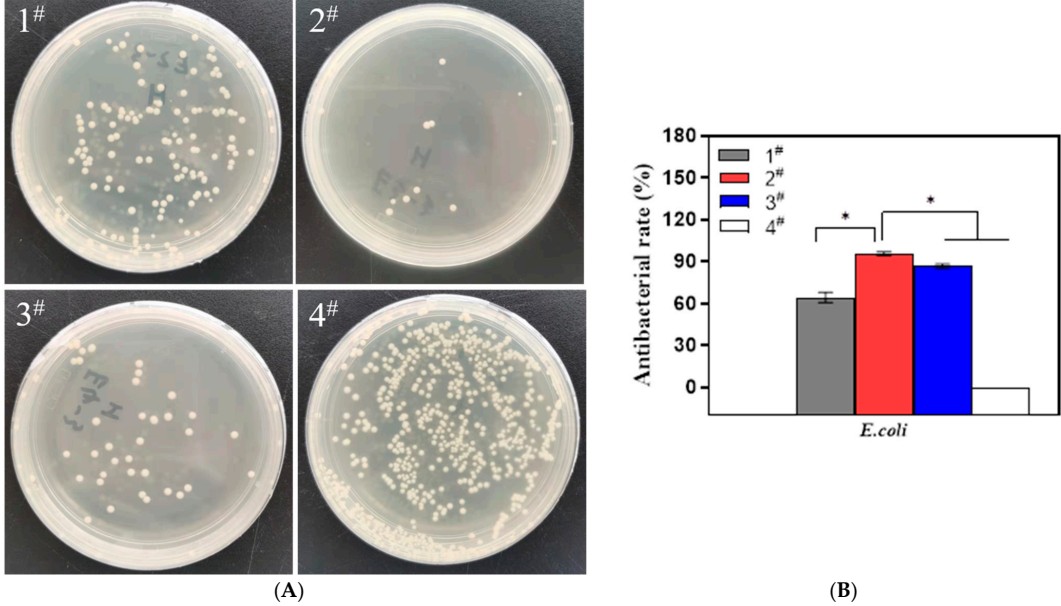

**Figure 7.** (**A**) Bacterial colonies on Ti-40Nb-10Ag for different sintering temperature (1#, 2#, 3#) and Ti-40Nb alloy (control group-4#); (**B**) Antibacterial rate of Ti-40Nb-10Ag for different sintering temperature (1#, 2#, 3#) and Ti-40Nb alloy (control group-4#). Error bars represent mean $\pm$ SD for n = 3, * $p < 0.01$.

Figure 7B shows the antibacterial rates of the Ti-40Nb alloy and Ti-40Nb-10Ag for different sintering temperatures. The antibacterial rate of 1# sample was only about 64.1%, while the antibacterial rate of 3# reached 86.5%, which suggests that these alloys were

not optimal antibacterial materials. The antibacterial rate of 4[#] sample was 0, which indicates that the alloy had no antibacterial properties at all. The 2[#] sample was prepared at 975 °C and its antibacterial property reached 95.6%, which indicates that the alloy had good antibacterial properties [32]. These results show that sintering temperature of Ti-40Nb-10Ag impacts the antibacterial property, which might relate to the change in microstructure.

Figure 8 shows the concentration of Ag element for different immersion time. It can be seen from the figure that the concentration of Ag element by the 2[#] sample was higher than other two groups, and the Ag element concentration of 3[#] sample was higher than 1[#] sample. This indicates that the antibacterial properties of the alloys are correlated to the different Ag element concentrations and are affected by the Ag concentration of the samples.

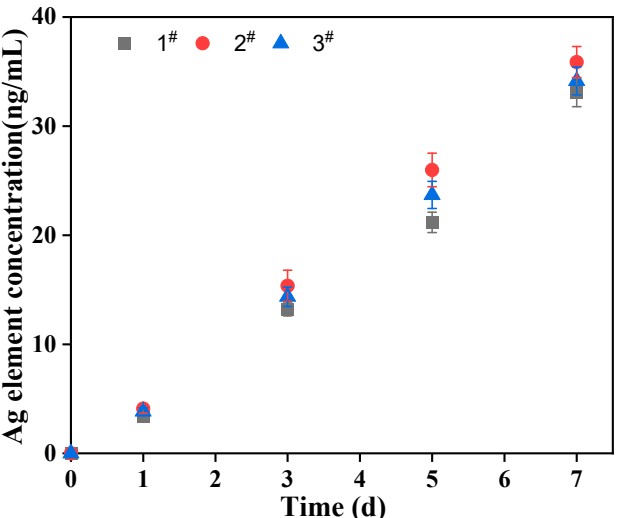

**Figure 8.** Concentration of Ag element for different immersion time in 0.9% NaCl solution.

This is because the $Ti_2Ag$ phase and the Ag-enriched area released positively charged Ag ions during the soaking process [33,34], while the bacterial surface biofilm exhibited negative charges. Therefore, when Ag metal ions make contact with microorganisms, the microbial protein structure would be destroyed, which results in the dysfunction and death of bacterial microorganisms [35–37]. When Ag ions make contact with the cell membrane of microorganisms, they are firmly combined to form an oligodynamic effect due to the negative charge in the cell membrane [38,39]. The oligodynamic effect led metal ions to penetrate the cell membrane and react with proteins [40]. This reaction would lead to protein coagulation and denaturation in microorganisms and destroy the activity of microbial synthases. Moreover, this reaction would interfere with the synthesis of microbial DNA and affect passage and reproduction, which induces microorganisms to lose the ability of division and proliferation. The combination of metal ions and proteins would damage the energy transmission system, material exchange system, and respiratory system of microorganisms [41].

## 4. Conclusions

According to the aforementioned results in assessing the influence of sintering temperatures of the Ti-40Nb-10Ag alloys for orthopedic replacement, the following conclusions can be drawn:

(1) Ti-40Nb-10Ag alloys were fabricated by spark plasma sintering technology with different sintering temperatures. The Ti-40Nb-10Ag alloys were composed mainly of α-Ti, β-Ti, and $Ti_2Ag$ intermetallic phases.

(2) The change in sintering temperatures can affect the microstructure of the alloy, which results in changes in microhardness, compressive strength, and elastic modulus. When the

sintering temperature was 975 °C, good metallurgical bonding was formed on the surface of the alloy with excellent microhardness, compressive strength, and elastic modulus.

(3)    The change in sintering temperature would affect the microstructure of the alloy, which results in the change in its antibacterial properties. When the sintering temperature was 975 °C, the alloy had excellent antibacterial ability and its antibacterial rate was 95.6%.

**Author Contributions:** Conceptualization, H.D. and Z.W.; methodology, B.Z.; software, B.Z.; validation, Y.C., B.Z. and W.W.; formal analysis, Y.Z.; investigation, P.Y.; resources B.Z.; data curation, B.Z.; writing—original draft preparation, B.Z.; writing—review and editing, H.D. and Z.W.; visualization, Y.Z.; supervision, Y.Z.; project administration, H.D.; funding acquisition, H.D. All authors have read and agreed to the published version of the manuscript.

**Funding:** This research was funded by the Youth Academic Project of Yunnan Province Basic Science Plan under grant No. 202101AU070110, the Joint Project of Kunming Medical University with Yunnan Province Science and Technology Department under grant No. 202201AY070001-085, Applied Basic Research Program of Yunnan Province under Grant No. 202001AU070040, and the Yunnan Province Clinical Center for Bone and joint Diseases under grant No. ZX2019-03-04.

**Institutional Review Board Statement:** Not applicable.

**Informed Consent Statement:** Not applicable.

**Data Availability Statement:** Not applicable.

**Conflicts of Interest:** The authors declare no conflict of interest.

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
