# Peer review of "Synthesis, Characterization and Antimicrobial Studies of Ti-40Nb-10Ag Implant Biomaterials"

_metals, doi:10.3390/met12081391_

Round 1

Reviewer 1 Report

1.       In Figure.3, Explain XRD peaks and interpret them with COD IDs.

Explain and mark the regions from SEM analysis in figure.4 for better understanding.

Reviewer 2 Report

Please describe the etching time for each sample. The etching time may significantly change the surface morphology of the sample.

In the Ag release test, does the 0.9% NaCl solution used as a solvent have a buffering effect? Normally, phosphate buffered saline would be used in most cases to maintain the pH at a value similar to that in vivo.

In Figure 3, the symbol of the β-Ti is shown at the sharp peak near 38° (the same position as α-Ti?) in 1#-4#, but the peak position is clearly different from that of Milled Powder, i.e. the peak shifts to higher diffraction angle. Please explain the reason for this phenomenon. The authors should also provide appropriate references (e.g., databases) for the peak positions.

In Figure 4, the authors explain that there is a difference from a viewpoint of pore formation. Was there a difference in surface roughness or surface area?

In Figures 5-7, please show the results of the statistical analyses between each condition. The reviewer thinks that a one-way ANOVA is one of the appropriate methods. This point is important to clarify significant differences between each condition.

In Figure 6, were the flexural strength and flexural modulus measured? Bending properties are also important, especially for use in orthopedic hip prostheses.

Figure 8 is a little confusing figure because the bars' positions of the 1# and 3 are off the regular abscissa positions. The reviewer thinks that it would be more appropriate to use a scatter plot. In addition, the reviewer recommends that the vertical axis not only shows the concentration of Ag but also what percentage of the Ag in the alloy was released.

Did you measure the changes in pH value in the Ag release test? If the solvent is not buffered, there may be a pH change.

When looking at Figure 9 in conjunction with the main text, it is considerably unclear what the figure is trying to explain because there are almost no appropriate words or phrases in the figure. A large modification of the figure is required.

In Figure 10, the description of contact angles and the scale bar are unclear. In Figure 10, there does not appear to be a significant difference in the contact angles for each condition. How many samples were measured for each condition? If the authors discuss the relationship between antimicrobial activity and hydrophilicity/hydrophobicity, it would be difficult to explain the above relationship without measuring multiple samples, calculating the average values and standard deviations, conducting suitable statistical analyses, and considering significant differences in contact angles for each condition.

Reviewer 3 Report

1-      In the introduction section please specify briefly about benefits of using SPS fabrication method.

2-      In section 2.1, Fig. 1, please remove images of Ti-40Nb and Ti-40Nb-10Ag powders as they do not add any information. Images of initial Ti, Nb and Ag powders are sufficient.

3-      In section 2.2, please remove the grit sizes (120#, 240#, 500#, 800#, 1000#, 1200# in sequence) and (200#, 320#, 400#, 600# and 1000#) from the text.

4-      In section 2.2, ‘Thatcherization’ should be modified to ‘characterization’.

5-      Section 2, Materials and Method, should be modified. Figure 2 and explanation about how the sintering temperatures were selected should be moved to the results and discussion section.

6-      From XRD patterns, please calculate the proportion of each of the phases at different temperatures.

Round 2

Reviewer 2 Report

The authors have responded to the questions from the reviewer well. Before the publication, please check the grrammatical errors and misspelling by the native English speakers.

Author Response

Thank you for the kind help, we have checked all grammatical errors and spelling in the revised manuscript.